# The Role of Long Noncoding RNAs in Intestinal Health and Diseases: A Focus on the Intestinal Barrier

**DOI:** 10.3390/biom13111674

**Published:** 2023-11-20

**Authors:** Qianying Lu, Yangfan Liang, Xiangyan Meng, Yanmei Zhao, Haojun Fan, Shike Hou

**Affiliations:** 1Institute of Disaster and Emergency Medicine, Tianjin University, Tianjin 300072, China; qianying.lu@tju.edu.cn (Q.L.); 2021435005@tju.edu.cn (Y.L.); mengxiangyan@tju.edu.cn (X.M.); houshike@tju.edu.cn (S.H.); 2Tianjin Key Laboratory of Disaster Medicine Technology, Tianjin 300072, China

**Keywords:** intestinal barrier, function, diseases, lncRNAs, biomarkers

## Abstract

The gut is the body’s largest immune organ, and the intestinal barrier prevents harmful substances such as bacteria and toxins from passing through the gastrointestinal mucosa. Intestinal barrier dysfunction is closely associated with various diseases. However, there are currently no FDA-approved therapies targeting the intestinal epithelial barriers. Long noncoding RNAs (lncRNAs), a class of RNA transcripts with a length of more than 200 nucleotides and no coding capacity, are essential for the development and regulation of a variety of biological processes and diseases. lncRNAs are involved in the intestinal barrier function and homeostasis maintenance. This article reviews the emerging role of lncRNAs in the intestinal barrier and highlights the potential applications of lncRNAs in the treatment of various intestinal diseases by reviewing the literature on cells, animal models, and clinical patients. The aim is to explore potential lncRNAs involved in the intestinal barrier and provide new ideas for the diagnosis and treatment of intestinal barrier damage-associated diseases in the clinical setting.

## 1. Introduction

The gastrointestinal mucosa is a surface where human interaction with the outside world occurs, and the intestinal barrier prevents harmful substances such as bacteria and toxins from passing through the gastrointestinal mucosa [1]. The intestinal barrier consists of many layers, including a chemical barrier, a physical barrier, an immune barrier, and a microbial barrier [2]. Under normal circumstances, each of the intestinal barriers performs its own functions to keep the body in a healthy state [3]. However, pathological injuries, such as sepsis, trauma, burn, ischemia/reperfusion (I/R), and radiation, lead to the overgrowth of pathogenic bacteria and intestinal barrier damage, which can lead to gastrointestinal disorders, the main symptoms of which are recurrent abdominal pain and bloating, heartburn, indigestion/dyspepsia, nausea and vomiting, diarrhea, and constipation [4]. In addition, intestinal barrier damage and increased vascular permeability allow pathogenic microorganisms to enter the bloodstream, causing systemic inflammation and even multiple organ failure [5,6]. Evidence suggests that intestinal barrier dysfunction is associated with a variety of diseases, such as inflammatory bowel disease (IBD), diabetes, nonalcoholic fatty liver disease, and psychiatric disorders, such as multiple sclerosis, Alzheimer’s disease, and Parkinson’s disease [7,8,9]. Intestinal barrier dysfunction has been considered a therapeutic target for many diseases. However, there are currently no FDA-approved therapies targeting epithelial barriers [10]. Therefore, a deeper understanding of the mechanisms underlying intestinal barrier regulation must be reached.

For a long time, research on the gene regulatory network focused on protein-coding genes. However, more recent studies have shown that less than 1.5% of genes in the human genome encode proteins, and most of them are transcribed into noncoding RNAs (ncRNAs) [11]. Long noncoding RNAs (lncRNAs) are a class of ncRNAs that are greater than 200 nucleotides. In the past, lncRNAs were considered to be “transcriptional noise” that lacked function [12]. In recent years, lncRNAs have been proven to play an important role in a variety of biological processes, such as cell cycle control, differentiation, apoptosis, cellular senescence, and immune responses, through chromatin modification, transcriptional regulation, and posttranscriptional regulation [13,14]. Due to their high tissue-specific and condition-specific expression, lncRNAs are considered to be potential biomarkers of various diseases [15,16]. In terms of intestinal barrier function, lncRNA can maintain intestinal homeostasis through various aspects, such as regulating intestinal epithelial regeneration, intestinal immunity, and intestinal flora [17,18,19].

In this article, we summarize the role of lncRNAs in the intestinal barrier and intestinal diseases by reviewing the literature on cells, animal models, and clinical patients. The aim is to explore potential lncRNAs as therapeutic targets for intestinal barrier dysfunction and provide new ideas for the clinical treatment of intestinal diseases.

## 2. Biogenesis and Regulatory Functions of lncRNAs

Most lncRNAs, which have 5′-end m7G caps and 3′-end poly(A) tails, are transcribed by RNA polymerase II (Pol II), and a small proportion of lncRNAs are transcribed by other RNA polymerases [20]. Unlike mRNAs, lncRNAs are weakly transcriptionally spliced, resulting in the accumulation of a small fraction of chromatin, and their subsequent degradation by exosomes in the nucleus, eventually leading to this fraction of lncRNAs remaining in the nucleus [21] (Figure 1A). These remaining fractions of lncRNAs can bind to chromatin, histones, and gene loci in the nucleus, thereby regulating chromatin structure and function as well as nearby or distant gene transcription. In addition, most lncRNAs are exported into the cytoplasm via nuclear RNA export factor 1 (NXF1) in a process similar to the process for mRNAs [22]. Upon reaching the cytoplasmic matrix, lncRNAs can interact with different RNA-binding proteins (RBPs), ribosomes, miRNAs, mitochondria, and other organelles, thereby affecting the splicing and degradation of target genes and organelle homeostasis [23,24] (Figure 1C).

LncRNAs have shown powerful and unique regulatory functions in the body; as important regulators of gene expression networks, the role of lncRNAs can be seen in different regulatory mechanisms, including pretranscriptional, transcriptional, and posttranscriptional regulation [25]. First, lncRNAs take part in pretranscriptional regulation (Figure 1B). LncRNAs can interact with DNA to form an RNA/DNA hybrid or with chromatin-modifying enzymes, thereby affecting the chromatin modification state [26,27]. For example, the lncRNA ANRASSF1 is transcribed from the opposite strand of the RASSF2A gene and is named TIN and PIN RNAs, which form RNA/DNA hybrids by interacting with genomic DNA at the transcription sites and recruiting chromatin-modifying PRC2 complexes to the promoter region of the protein-coding RASSF1A gene, leading to histone H3K27 methylation at the RASSF3A promoter [26]. HOX transcript antisense intergenic RNA (HOTAIR), an epigenetic regulator of epidermal tissue development, is one of the most extensively investigated lncRNAs [28]. It interacts with polycomb repressive complex 2 (PRC2), a chromatin-associated methyltransferase that catalyzes the methylation of lysine 27 on histone H3 (H3K27), thus regulating the transcriptional silencing of genes of the HOXD locus and other gene loci [29,30].

Second, lncRNAs take part in transcriptional regulation (Figure 1B). LncRNAs can regulate the functions of promoters, enhancers, and transcription factors of target genes, thereby enhancing or inhibiting the expression of parental genes or other genes [23,31]. For example, lncRNA nuclear paraspeckle assembly transcript 1 (NEAT1) is a multiple endocrine neoplasia type 1 locus (Chr 11q13.1) transcript that is involved in familial tumor syndromes [32]. It is a structural lncRNA that participates in the constitution and assembly of nuclear paraspeckles [33]. Geng et al. [34] reported that lncRNA NEAT1 could bind to CCCTC-binding factor (CTCF), which can bind to the HOXA10 promoter and regulate histone modification levels, thus downregulating the expression of the HOXA10 gene. Therefore, lncRNA NEAT1 can affect promoter activity to regulate gene transcription. Moreover, eRNAs are ncRNAs transcribed bidirectionally at enhancers. Upon 17β-estradiol (E2)-bound estrogen receptor α (ERα) activation, the NRIP1 enhancer (eNRIP) is bidirectionally transcribed into eRNA and recruits cohesin to form chromatin loops, thereby facilitating the contact between the NRIP1 enhancer and the promoters of NRIP1 and trichostatin factor 1 (TFF1), which affects gene transcription [35].

Finally, lncRNAs are involved in posttranscriptional regulation. LncRNA mediates the cleavage and degradation of mRNAs by binding to proteins. In addition, lncRNAs can competitively bind to miRNAs, thereby affecting the regulation of mRNAs via miRNAs [23,31,36]. For example, lncRNA H19 has been reported to take part in various pathological processes through various mechanisms [37,38]. In intestinal pathologies, H19 functions as a sponge for miRNAs, such as let-7g, miR-34a, and miR-874, thus affecting the intestinal barrier function via epithelial regeneration-related genes [39,40].

## 3. The Emerging Role of lncRNAs in Intestinal Barrier

The intestinal barrier, which consists of many layers, including a chemical barrier, a physical barrier, an immune barrier, and a microbial barrier, is important for maintaining intestinal homeostasis and body health. Since intestinal barrier dysfunction can lead to a variety of diseases, it is very important to investigate the relevant mechanisms and regulatory strategies of intestinal barrier dysfunction. To date, studies have shown that many lncRNAs are involved in the intestinal barrier’s homeostasis maintenance (Figure 2).

### 3.1. LncRNAs in Intestinal Chemical Barrier

The intestinal chemical barrier is mainly composed of mucus layers, which are located at the interface between the luminal microflora and the epithelium. The mucus layer is a layer of gel that consists of water, electrolytes, lipids, proteins, and cell debris [41]. Among its components, mucins, a class of specific mucus proteins that can bind to water and impart mucinous gel-like properties, are the major structural and functional components. There are two types of mucins: transmembrane mucins and gel-forming mucins [42]. Transmembrane mucins are synthesized and attached to the cell membrane of intestinal cells, whose function is mainly cell protection. Gel-forming mucins are secreted from goblet cells and make up the skeleton of the mucus layer [43]. After synthesis and secretion, they expand and form a stratified mucus gel, which serves as a physical barrier and a hydrophobic barrier to block the entry of bacteria [44]. In addition, a variety of secreted immune factors, such as antimicrobial peptides (AMPs) and secretory immunoglobulin a (sIgA), also provide a powerful weapon to prevent infection by inhibiting intestinal pathogenic microorganisms and preventing intestinal microorganisms and their toxin molecules from attacking the intestinal mucosa [45].

Studies have shown that lncRNAs participate in the intestinal chemical barrier, mainly in mucus production (Figure 2B). For example, studies have shown that the high expression of H19 in pathological conditions is closely associated with intestinal barrier dysfunction. In a previous study, compared with intestinal organoids from control littermate mice, intestinal organoids isolated from H19−/− mice had increased numbers of lysozyme and mucin 2-positive cells and showed increased tolerance to LPS, which suggests an inhibition effect of lncRNA H19 in mucus production [46]. There are few studies on the role of lncRNAs in the intestinal chemical barrier, and more in-depth studies are needed.

### 3.2. LncRNAs in Intestinal Physical Barrier

The intestinal mucosal epithelial cells, cell–cell junctions, and the basement membrane constitute the intestinal epithelial physical barrier [47]. This intestinal epithelium is composed of a large number of repetitive, continuously self-renewing crypt-villus units [48], including various types of intestinal epithelial cells. These epithelial cells are tightly packed, like walls, to protect against pathogenic invasion, and different cells have different functions. Intestinal stem cells are considered the source of all intestinal cells and are responsible for epithelial renewal in physiological conditions and intestinal regeneration in response to injury [49]. Stem cells divide and produce transit-amplifying cells (TA cells), which then differentiate downward into Paneth cells and upward into goblet cells, enteroendocrine cells, and intestinal absorptive cells [50]. Paneth cells are located in the crypt of the small intestine, and they can produce antimicrobial peptides and immunomodulatory proteins to regulate the composition of intestinal bacteria [51]. Goblet cells are characteristic of the intestinal epithelium, and their most important function is to secrete mucus, as mentioned above [52]. Enteroendocrine cells can produce and release hormones/signaling molecules and, thus, modulate a variety of physiological gastrointestinal, and homeostatic functions [53]. The function of intestinal absorptive cells is to absorb nutrients from food. In tissues, these cells are connected using cell–cell junctions, including tight junctions (TJs), adherens junctions (AJs), and desmosomes [54]. Among them, tight junctions are the core. Tight junction molecules consist of ZO protein, Occludin, claudins, and junctional adhesion molecules (JAMs), and they regulate the paracellular permeability of water, ions, and macromolecules in adjacent cells [55]. Once the tight junctions of intestinal epithelial cells are mutated, reduced, or lost, the permeability of the intercellular space will increase, and bacteria, endotoxins, and macromolecules can enter the systemic circulation [56]. The intestinal physical barrier is a selective permeability barrier that allows the absorption of nutrients, electrolytes, and water and defends against the invasion of pathogenic microorganisms [57]. Epithelial cell injury and cell–cell junction damage will lead to intestinal epithelial barrier disruption; therefore, investigating and discovering lncRNAs that can regulate epithelial cell injury and cell–cell junction damage is of importance.

Studies have shown that lncRNAs play a crucial role in intestinal epithelial cells (Figure 2C). For example, it has been shown that WiNTRLINC1 interacts with TCF4/β-catenin to positively regulate the expression of ASCL2, thereby maintaining the stemness of intestinal stem cells [58,59]. Similarly, the long noncoding RNA Gata6 (lncGata6) is also essential for the maintenance of intestinal stem cell (ISC) stemness as well as epithelial regeneration. Mechanistically, it recruits the NURF complex to the Ehf promoter to induce its transcription, thereby promoting the expression of Lgr4/5 to enhance the activation of Wnt signaling [60]. In addition to ISCs, lncRNAs can also affect intestinal barrier function by affecting other intestinal epithelial cells. For example, H19 overexpression inhibits autophagy and the number of goblet and Paneth cells, thereby disrupting the intestinal barrier, while H19 gene knockdown promotes barrier function to cope with septic stress [46]. In a previous study, lncRNA uc.173 was reported to be involved in intestinal mucosal growth, and increasing levels of uc.173 increased the growth of intestinal epithelial cells and organoids. Uc.173 specifically inhibits miR-195 expression, which is crucial for controlling intestinal epithelial cell (IEC) proliferation [61].

Moreover, studies have reported that various lncRNAs are involved in the maintenance of intestinal barrier integrity by affecting cell–cell junctions. For example, a study has shown that lncRNA H19 acts as a precursor of miR-675, and overexpression of H19 increases the expression of miR-675, thereby disrupting the translation of the mRNA-encoding tight junction proteins E-cadherin and Zo-1, leading to epithelial barrier dysfunction [62]. Uc.173 can enhance epithelial barrier function by interacting with miR-29b to promote the translation of TJ claudin-1 (CLDN1) [63]. Xiao et al. [64] reported that lncRNA SPRY4-IT1, a 706-base pair transcript, can directly interact with mRNA-encoding TJ proteins, including claudin-1, claudin-3, JAM-1, and Occludin, thus regulating the intestinal barrier. And deletion of SPRY4-IT1 leads to epithelial barrier dysfunction.

Therefore, lncRNAs can affect intestinal epithelial cells and cell–cell junctions, thus regulating the intestinal physical barrier.

### 3.3. LncRNAs in Intestinal Immune Barrier

As the intestine is the largest immune organ in the body, the intestine plays a vital role in the maintenance of immune homeostasis. The intestinal immune system, which recognizes and responds differently to the flora and antigens in the intestinal tract to maintain the immune balance of the intestinal tract and the whole body, is composed of gut-associated lymphoid tissue (GALT) and secreted immunoglobulins [65]. The GALT of humans and mice is composed of multi-follicular lymphoid tissues and numerous isolated lymphoid follicles (ILF), mainly including immune cells such as T cells, B cells, dendritic cells (DCs), mast cells, and macrophages [66,67]. Secretory immunoglobulin A (sIgA), a protein composed of two identical heavy (H) and light (L) chains connected by disulfide bonds, is the most highly secreted immunoglobulin in the gut [68]. The intestinal immune system consists of two parts: an innate immunity system and an adaptive immunity system. The innate immune system, which includes macrophages, DCs, and natural killer (NK) cells, recognizes microbial polysaccharides, glycolipids, lipoproteins, nucleotides, and nucleic acids; it provides the first line of defense against common microbes [69,70]. The adaptive immune system includes T and B lymphocytes and induces specific memory responses to certain antigens [71]. B cell production of IgA has been shown to have a significant influence in mediating mucosal immunity as the first part of the defense mechanism. The intestinal immune system plays a pivotal role in the maintenance of intestinal homeostasis, and its imbalance is closely related to intestinal diseases [72]. And lncRNAs can affect intestinal immune balance by regulating the functions of the intestinal innate immunity and adaptive immunity systems.

LncRNAs can regulate innate immunity (Figure 2C). For example, lncRNAs affect intestinal macrophage polarization. It has been demonstrated that NEAT1 can regulate the intestinal epithelial barrier and macrophage polarization, thereby affecting intestinal inflammation. Downregulation or knockdown of NEAT1 promotes macrophage polarization to M2, thereby inhibiting inflammatory effects [73]. In LPS-treated peritoneal macrophages, the expression of lncRNA-Cox2 has been observed to be upregulated, whereas silencing lncRNA-Cox2 reduces LPS-induced expression of TNF-α, IL-6, and IL-1β in the cell supernatant by hindering LPS-induced M1 macrophage polarization [74]. In addition, lncRNAs affect the expression of cytokines. In intestinal epithelial cell lines and macrophages, the expression of transmembrane channel-like 3 (TMC3)-AS1 is elevated following LPS stimulation. TMC3-AS1 negatively regulates the expression of IL-10 by preventing p65 from binding to the IL-10 promoter to modulate the innate immune response [75]. Feng et al. [76] reported that lncRNA MALAT1 was highly expressed in mice with food allergy (FA). Its silencing could reduce the severity of FA by decreasing the secretion of IL-6 by DCs and suppressing the immunomodulation of Tregs.

LncRNAs are also involved in the function of adaptive immunity. For example, lncRNA IFNG-AS1 is located on chromosome 12 and is in close proximity to the inflammatory cytokine interferon-γ (IFNG). IFNG-AS1 is upregulated in ulcerative colitis patients to regulate the key inflammatory cytokine IFNG in CD4 T cells [77]. Nie et al. [78] analyzed the lncRNA expression profiles of IBD patients and found that lnc-ITSN1-2 was significantly upregulated. It can promote CD4 T cell activation, proliferation, and Th1/Th17 cell differentiation through the lnc-ITSN1-2/miR-125a/IL-23R axis. The lncRNA DQ786243 can affect CREB and Foxp3 expression and, thus, regulate the function of Tregs, which maintain immune homeostasis and suppress inflammation [79].

### 3.4. LncRNAs in Gut Microbial Barrier

The gastrointestinal tract is rich in microbes, most of which are bacteria, mainly *Bacteroides*,* Firmicutes*,* Proteobacteria*,** and *Actinomycetes* [80]. The commensal intestinal microbiome on the mucosal surface of the intestinal epithelium constitutes the intestinal microbial barrier, which is one of the important components of the intestinal barrier [47]. Studies have shown that intestinal microbes play important roles in their host [81]. The gut microbiome provides energy and nutrients to the host. In humans, members of the gut microbiome have been shown to synthesize short-chain fatty acids (SCFAs), vitamin K, and most water-soluble B vitamins. For example, *Bifidobacteria* and *Lactobacilli* biosynthesize the B-group vitamin folate, and *Bacteroides* biosynthesize vitamin K [82,83]. The gut microbiota ferments indigestible complex carbohydrates and proteins to produce SCFAs, which play an important role in regulating energy metabolism and energy supply, as well as maintaining host homeostasis [81,84]. Intestinal microbes also play an important role in maintaining intestinal barrier homeostasis, such as maintaining mucus layer thickness and affecting tight junction protein expression [85]. In addition, intestinal microbes also assist in the development of the host immune system by regulating the proliferation and activation of immune cells, as well as cytokine production and other mechanisms [86]. The gut microbiota is critical to human health, and it is necessary to improve the balance of the intestinal microbiota.

Evidence suggests that there is an interconnected link between lncRNAs and gut microbial balance, mainly showing the effect of the gut microbiota on lncRNAs (Figure 2A). For example, Cui et al. [87] found that the gut microbiota of mice was altered after irradiation. Fecal microbiota transplantation preserved the intestinal bacterial composition and mRNA and lncRNA expression profiles. Liang et al. [88] reported that the expression profiles of lncRNAs in the intestinal tissues exhibited molecular characteristics that reflect the microbial types. They found that the lncRNA expression profiles were different between germ-free, conventional, and different gnotobiotic mice, and the prediction model based on lncRNAs successfully recognized these different germ-bearing mice. Their findings indicated the important role of lncRNAs in host–microbe interactions. Yang et al. [19] reported that in an *M. tuberculosis* (TB)-infected mouse model, gut dysbiosis destroyed pulmonary homeostasis and worsened TB infections. LncRNA-CGB, a lncRNA regulated by intestinal symbiotic bacteria, was downregulated due to intestinal microbiota imbalance during TB infection. It could induce the production of IFN-γ to mediate a reduction in host sensitivity to TB caused by gut microbes. Lin et al. [89] reported that lncRNA ENO1-IT1, an important target of *F. nucleatum*, could regulate inflammatory response via the ENO1-IT1/miR-22-3p/IRF5 axis in an *F. nucleatum*-induced NEC-inflammation model. These results suggest that the intestinal flora influences lncRNAs in a variety of intestinal injury models.

## 4. The Role of lncRNAs in Intestinal Diseases

### 4.1. LncRNAs and IBD

IBD is a chronic inflammation disease of the intestine that encompasses Crohn’s disease and ulcerative colitis, with patients showing symptoms such as diarrhea, abdominal pain, rectal bleeding, and weight loss [90]. Researchers have reported that the intestinal physical barrier function plays an important role in the pathogenesis of IBD (Table 1). For example, in patients with IBD, epithelial cell apoptosis has been found to be increased, while the expression of tight junction proteins is decreased, which indicates damage to the intestinal physical barrier in IBD patients [91,92]. LncRNAs can affect the intestinal physical barrier in IBD. On the one hand, lncRNAs regulate epithelial cell apoptosis in IBD. A previous study found that LncRNA KIF9-AS1 expression was upregulated in IBD patients, DSS-induced IBD mice, and DSS-induced colonic cells. In contrast, lncRNA KIF9-AS1 silencing inhibited the apoptosis of DSS-induced colonic cells via the miR-148a-3p/SOCS3 axis, thus alleviating colon injury and inflammation in IBD patients [93]. Similarly, lncRNA CRNDE promotes DSS-induced apoptosis in colonic epithelial cells by increasing SOCS1 through binding to miR-495, which aggravates the IBD process [94]. In addition, Wu et al. [95] found that BC012900 was significantly increased in patients with active UC, and BC012900 overexpression resulted in the inhibition of cell proliferation and increased cell susceptibility to apoptosis. On the other hand, lncRNAs affect intestinal tight junction proteins in IBD. For instance, lncRNA NEAT1 affects the inflammatory response of the intestinal barrier by regulating the expression of ZO-1 and Occludin. LncRNA PlncRNA1 binds to miR-34c to regulate the expression of Myc-associated zinc finger protein (MAZ), ZO-1, and Occludin [96]. In addition, lncRNAs regulate the intestinal physical barrier via other mechanisms. GAS5 affects the intestinal barrier of IBD patients by regulating the expression of MMP9 and MMP2 [97]. LncRNA PSCK6 exacerbates the mucosal barrier damage associated with chronic colitis via the HIPK1-STAT1 axis [98]. And lncRNA CCAT1 destroys the intestinal barrier through the miR-185-3p/MLCK signaling pathway [99].

Moreover, environmental, genetic, and microbiological factors interact with the immune system, resulting in a dysregulated immune response that leads to the development of IBD [100]. lncRNAs can affect the immune response in IBD. Lnc78583 mediates the miR-3202/HOXB13 signaling pathway by inhibiting the expression of inflammatory factors [101]. IFNG-AS1 stimulates the production of inflammatory factors by influencing the expression of IFNG and IL2 [102]. DQ786243 regulates the expression of CREB and Foxp3, which then affects the function of Tregs, a class of T cells that maintain immune homeostasis and limit inflammation [79]. LncRNA ITSN1-2 (lnc-ITSN1-2) leads to increased inflammation by positively regulating IL-23R and CD4 T cell activation and proliferation and Th1/Th17 cell differentiation [78]. NEAT1 and NAIL also influence the inflammatory response in colitis by influencing the NF-κB signaling pathway [103,104].

Therefore, lncRNAs can affect the intestinal physical barrier and immune response in IBD, which may provide new therapeutic targets for IBD.

**Table 1 biomolecules-13-01674-t001:** LncRNAs and IBD.

Long Noncoding RNA	Expression	Models	Functions	Potential Mechanism	References
KIF9-AS1	↑	C57BL/6 mice, HT-29, UC patient’s colon	apoptosis	miR-148a-3p/SOCS3	[93]
CRNDE	↑	C57BL/6 mice, HT-29, LOVO, Caco-2	apoptosis	miR-495/SOCS1	[94]
BC012900	↑	IBD patient’s colon tissue, HT29, Caco-2, HCT116	apoptosis	PPM1A	[95]
PlncRNA1	↑	Caco-2	intestinal epithelial barrier	miR-34c-MAZ, ZO-1 and Occludin	[96]
GAS5	↓	THP1, IBD patient’s colon	intestinal epithelial barrier	MMP2 and MMP9	[97]
PCSK6-AS1	↑	IBD patient’s colon tissue, C57BL/6 mice	inflammation	HIPK1-STAT1	[98]
CCAT1	↑	IBD patient’s colon, Caco-2	intestinal epithelial barrier	miR-185-3p/MLCK	[99]
NEAT1	↑	RAW264.7, C57BL/6 mice	intestinal epithelial barrier	modulating the inflammation	[73]
	↑	NCM460, HT-29, C57BL/6 mice	inflammation	TNFRSF1B or NF-κB pathway	[103]
	↑	BALB/c mice	PDT	miR-204-5p-PI3K-Akt axis	[105]
lnc78583	↑	FHCs, IBD patient’s colon	inflammation	miR-3202-HOXB13	[101]
IFNG-AS1		UC patient’s colon, Jurkat cells	inflammation	IFNG, IL2	[102]
DQ786243	↑	CD patient’s colon, Jurkat cells	Inflammation	CREB and Foxp3 affect Treg.	[79]
lnc-ITSN120-60	↑	IBD patient’s colon tissue	inflammation	miR-125a-IL-23R-CD4 T cell	[78]
NAIL	↑	BMDM	inflammation	p38, NFκB	[104]

CD: Crohn’s disease; SOCS3: suppressor of cytokine signaling; Caco-2: human colorectal cancer cell line; HT-29: human colon cancer cell line; SOCS1: suppressor of cytokine signaling; PPM1A: protein phosphatase; NCM460: human normal colonic epithelial cell line; TNFRSF1B: tumor necrosis factor superfamily member 1B; PDT: photodynamic therapy; ZO-1: zonula occludens 1; MAZ: Myc-associated zinc finger protein; THP1: human monocytic cell line; MMP2: matrix metallopeptidase 2; MLCK: myosin light-chain kinase; HOXB13: homeobox B13; IFNG: inflammatory cytokines interferon-γ; CREB: CAMP response element binding protein; Foxp3: forkhead box P3; BMDM: bone marrow-derived macrophages. “↑” arrows represent expressions going up, “↓” arrows represents expressions going down.

### 4.2. LncRNAs and Irritable Bowel Syndrome (IBS)

IBS is a complex gastrointestinal disorder characterized by recurrent abdominal pain, bloating, and changes in stool characteristics. However, the pathogenesis of IBS is still unclear [106]. It has been reported IBS is caused by enhanced permeability of the intestinal mucosa, which is associated with low expression levels of aquaporins (AQPs) 1, 3, and 8 [107]. LncRNA H19 overexpression induces a rise in AQP1 and 3 expression levels, thereby affecting the intestinal barrier [108]. In addition, visceral hypersensitivity is one of the causes of IBS. LncRNA XIST plays a regulatory role in 5-HT-induced visceral hypersensitivity in mice with diarrhea-predominant IBS by recruiting the methylases DNMT1, DNMT3A, and DNMT3B to facilitate SERT promoter methylation and reduce its expression [109]. Therefore, these research studies indicate that lncRNAs play regulatory roles in IBS (Table 2).

### 4.3. LncRNAs and Radiation-Induced Intestinal Injury

Radiation-induced intestinal injury (RIII) is one of the main side effects caused by radiation therapy, which can cause patients to develop symptoms such as anorexia, vomiting, diarrhea, systemic infections, and in more serious cases, even sepsis or death. It affects treatment outcomes, reduces patients’ quality of life, and brings great inconvenience to patients with abdominal or pelvic tumors [110]. As radiotherapy is one of the traditional methods of tumor treatment, its side effects have attracted increasing attention. However, there are currently no FDA-approved drugs for the treatment of radiation-induced intestinal injury. Our previous study showed that the intestinal lncRNAs of mice changed after irradiation via whole transcriptome sequencing, suggesting that lncRNAs may act during the process of RIII [111].

There is evidence that the pathophysiology of RIII is correlated with intestinal microbiota dysregulation, and bacteria-associated sepsis in the gastrointestinal tract is a serious problem for patients with RIII [112]. Cui et al. [87] found that the gut microbiota of mice was altered after irradiation. Fecal microbiota transplantation affected the expression profiles of mRNAs and lncRNAs of the host to reduce radiation-induced toxicity. LncRNA GAS5 can silence CCL1 gene transcription, which is an essential cytokine for M2bMϕ survival. The expression of lncRNA GAS5 in macrophages at the bacterial translocation sites decreased in irradiated mice—glycyrrhizin could improve GAS5 reduction, thus protecting these irradiated mice from gut microbiome imbalance-induced intestinal injury [113]. Therefore, the research findings described above showed that bacterial imbalance could cause changes in lncRNA expression, and altering lncRNA expression could improve intestinal injury caused by intestinal flora imbalance in a radiation-induced intestinal injury model.

In addition, in patients suffering from radiation-induced intestinal injury, the number of inflammatory cells increases, and infiltration of lymphocytes and neutrophils in the intestines occurs, which suggests that radiation disrupts the immune balance of the intestines. And lncRNAs are involved in radiation-induced fibrosis, a chronic injury caused by persistent inflammation. Zhou et al. [114] found that lncRNA WWC2-AS1 could interact with miR-16 and WWC2-AS1 overexpression decreased the expression of miR-16, thereby contributing to the expression of FGF2 and leading to intestinal fibrosis.

### 4.4. LncRNAs and Colon Cancer

Colon cancer is one of the leading tumors in the world and is considered one of the most prevalent cancers, along with lung, prostate, and breast cancers. Various lncRNAs promote the development and progression of colon cancer (Table 3).

First, lncRNA affects colon cancer by influencing cell proliferation. OIP5-AS1 is highly expressed in colon cancer, and silencing OIP5-AS1 inhibits the malignant behavior of colon cancer [115]. ELFN1-AS1 is upregulated in colon cancer cells via hypoxia-induced upregulation, and it competitively and endogenously binds to miR-191-5p to upregulate the tripartite motif TRIM 14 (TRIM14) to promote colon cancer growth and metastasis [116]. DANCR is highly expressed in colon cancer tissues and cell lines, and silencing DANCR inhibits the proliferation, survival, and metastasis of colon cancer via the DANCR/miR-518a-3p/MDM2 axis [117]. Small nucleolar RNA host gene 16 (SNHG16) overexpression leads to the downregulation of miR-302a-3p expression via promoting AKT expression to induce the proliferation of colon cancer cells, thus participating in the growth of colon cancer [118].

Secondly, lncRNA affects colon cancer by influencing cell migration, metastasis, and invasion. For example, vascular endothelial growth factor A (VEGFA) plays a major role in promoting colonic metastasis through Sox2-associated signaling. And lncRNA plasmacytoma variant translocation 1 (PVT1) promotes the expression of VEGFA and epidermal growth factor receptor (EGFR) by negatively regulating miR-152-3p [119]. DSCAM-AS1 is significantly upregulated in colon cancer, and a high expression level of DSCAM-AS1 is associated with poor prognosis in colon cancer patients. DSCAM-AS1 acts as an oncogenic lncRNA by regulating the miR-204/SOX4 axis and, thus, regulating the migration of cancer cells [120]. LINC00941 can regulate MYC expression by directly interacting with miR-205-5p, and LINC00941 overexpression promotes the proliferation, migration, and invasion ability of colon cancer cells [121]. LINC-PINT inhibits tumor cell invasion to suppress cancer progression. LncRNA AGAP2-AS1 overexpression and LINC-PINT form a negative feedback loop on colon cancer cell proliferation, migration, and invasion [122]. The expression of LncRNA NEAT1 is closely related to the overall survival of colon cancer patients, and overexpression of NEAT1 can significantly promote the migration and invasion of colon cancer cells by competitively and endogenously binding to miR-185-5p to regulate IGF2 [123]. Thirdly, lncRNAs affect colon cancer by influencing cell stemness. LINC01606 promotes SCD1 expression by interacting with miR-423-5p, thereby controlling the synthesis of intracellular MUFAs by activating Wnt/β-catenin protein signaling to promote cancer cell stemness [124]. B4GALT1-AS1 promotes yes-associated protein (YAP) transcriptional activity, thus mediating stemness in colon cancer cells [125]. Competitive endogenous binding of the lncRNA solute carrier organic anion transporter family member 4A1-AS1 (SLCO4A1-AS1) to miR-150-3p inhibits SLCO150A1 expression, thereby inhibiting the development of colon cancer stem cells and preventing colon cancer progression [126].

Finally, lncRNAs can affect the treatment of colon cancer. For example, lncRNAs are involved in drug resistance [127]. LncRNA-PCAT1 is highly upregulated in colon cancer tissues and cancer cell lines, and inhibition of lncRNA-PCAT1 expression can enhance the sensitivity of colon cancer cells to chemotherapy [128]. Oxaliplatin resistance presents a major obstacle in the treatment of locally advanced and metastatic colon cancer. Lnc-RP11-536 K7.3 induces SOX2 transcription to activate USP7 mRNA expression, thereby promoting oxaliplatin resistance [129]. CCAT1 is significantly upregulated in both colon cancer tissues and tumor cells. Inhibition of CCAT1 expression can significantly enhance tumor sensitivity to 5-FU. Further studies have shown that CCAT1 promotes cell proliferation and enhances drug resistance by regulating the miR-143/PLK1/BUBR1 signaling axis [130]. In addition, lncRNAs also affect radiotherapy. lncRNA TTN-AS1 is upregulated in colorectal cancer cells after receiving radiotherapy. Knockdown of TTN-AS1 increases the radiation sensitivity of colorectal cancer cells by inhibiting miR-134-5p and, thus, increasing PAK3 expression, which may be related to the P21 and AKT/GSK-3β/β-catenin pathways [131].

Therefore, lncRNAs are involved in colon cancer, which may open up a new way for the treatment of colon cancer.

**Table 3 biomolecules-13-01674-t003:** LncRNAs and colon cancer.

Long Noncoding RNA	Expression	Models	Tumor Suppressor/Oncogenic	Functions	Potential Mechanism	References
OIP5-AS1	↑	CC patient’s tumor, SW620, HT-29, HCT116, LoVo, RKO, NCM460	oncogenic	growth	miR-34b-5p- HuR-PI3K/Akt	[115]
ELFN1-AS1	↑	HCT116, SW480, LoVo, HT29	oncogenic	growth and metastasis	miR-191-5p—TRIM14	[116]
DANCR	↑	CC patient’s tumor, HT29, HCT116, SW116, Caco-2	oncogenic	growth	miR-518a-3p/MDM2	[117]
SNHG16	↑	HCT116, CaCO-2	oncogenic	growth	miR-302a-3p/AKT axis	[118]
PVT1	↑	CC patient’s tumor, SCID mice, HCT116	oncogenic	metastasis	EGFR and VEGFA	[119]
DSCAM-AS1	↑	CC patient’s tumor, HT29, HCT8, SW480, LOVO	oncogenic	proliferation and migration	miR-204/SOX4 axis	[120]
LINC00941	↑	CC patient tumor, LOVO, HCT116	oncogenic	proliferation and invasion	miR-205-5p-MYC	[121]
AGAP2-AS1	↑	CC patient’s tumor	oncogenic	proliferation, migration, and Invasion	LINC-PINT	[122]
NEAT1	↑	CC patient’s tumor, SW620 HT-29, HCT 116, LoVo, and SW480, NCM460	oncogenic	invasion and migration	miR-185-5p/IGF2 axis	[123]
SNHG12	↑	CC patient’s tumor, BALB/c nude mice, Lovo, HCT116, SW480 and HT29, HIEC	oncogenic	development and progression	miR-15a-PDK4 axis	[132]
LINC01606	↑	COAD tissue, SW480, HT29, HEK293T	oncogenic	stemness and ferroptosis resistance	SCD1-Wnt/β- catenin-TFE3	[124]
B4GALT1-AS1	↑	HCT-116, SW480,SW620, HT-29, CT-26, SW1116, NCM460	oncogenic	stemness and migration	YAP	[125]
SLCO4A1-AS1	↑	HCT116, NCM460	oncogenic	migration, invasion, spheroidization, and tumor formation	miR-150-3p -SLCO4A1	[126]
LncRNA-PCAT1	↑	CC patient’s tumor; HCT116	oncogenic	chemoresistance	Bax/Bcl-2	[128]
Lnc-RP11-536 K7.3	↑	CC patient’s tumor	oncogenic	proliferation, glycolysis, angiogenesis	SOX2/USP7/HIF-1α	[129]
CCAT1	↑	ESCC cell lines	oncogenic	chemoresistance	miR-143/PLK1/BUBR1	[99]
TTN-AS1	↑	SW620 and HT29	oncogenic	radiation sensitivity	miR-134-5p/PAK3	[131]

CC: colon cancer; TRIM14: tripartite motif TRIM 14; SCID: severe combined immune deficiency; EGFR: epidermal growth factor receptor; VEGFA: vascular endothelial growth factor A; SCD1: stearoyl-CoA desaturase 1; PDK4: pyruvate dehydrogenase kinase 4; TFE3: IGHM enhancer 3; YAP: yes-associated protein. “↑” arrows represent expressions going up, “↓” arrows represents expressions going down.

## 5. Conclusions

The gut is an important digestive organ of the human body, and the intestinal barrier is associated with a variety of diseases and may be considered a therapeutic target [10]. However, there are currently no FDA-approved therapies targeting epithelial barriers. Therefore, a deeper understanding of the mechanisms underlying intestinal barrier regulation must be reached. This article reviews the emerging role of lncRNAs in the intestinal barrier and potential treatment strategies for diseases related to intestinal barrier dysfunction to provide new ideas for the clinical treatment of intestinal diseases.

Since research on lncRNAs has only recently gained traction, knowledge regarding the role of lncRNAs in the intestinal barrier is still lacking. Let us take the gut microbiome imbalance as an example—although the intestinal microbiota has been studied for a long time regarding the progression of intestinal diseases [133,134,135], studies on intestinal microbiota–lncRNA interactions have been carried out only in recent years. Moreover, these studies are more focused on the influence of the intestinal microbiota on lncRNAs. There are few studies of the influence of lncRNAs on intestinal microbiota imbalance. Since microbial imbalance affects intestinal barrier dysfunction, understanding which lncRNAs play a role in this process is crucial for the early control of disease development.

Moreover, in this review, we show that lncRNAs play important roles in the intestinal barrier and intestinal diseases, such as IBD, IBS, colon cancer, and radiation-induced intestinal injury. LncRNAs have an impact on these disease processes by affecting the intestinal barrier. For example, NEAT1 is highly expressed in IBD patients and participates in the inflammatory response by regulating the intestinal epithelial barrier [73]. DANCR may affect colon cancer cell growth and metastasis through the miR-518a-3p/MDM2 axis [117]. Thus, these lncRNAs are promising markers for diagnosis and treatment of intestinal barrier-related diseases. In addition, researchers have developed techniques targeting lncRNAs, which provides conditions for the application of lncRNAs. For example, Naveed et al. developed a targeted ASO approach to sterically block NEAT1_1 polyadenylation processing to combat neuroblastoma [136]. Vaidya et al. demonstrated the effectiveness of a triple-negative breast cancer therapy by targeting nanoparticle-mediated RNAi of the oncogenic lncRNA DANCR [137]. These are all potential strategies for cancer treatment.

However, there are still many challenges to overcome. First, we comprehensively reviewed the role of lncRNAs in the intestinal barrier and intestinal diseases. However, due to the wide variety of lncRNAs and their extensive roles, specific lncRNAs still need to be explored further. Second, although there are existing studies showing that lncRNAs are involved in the occurrence and development of intestinal diseases, these studies have mostly focused on colon cancer and IBD, and there are few studies on radiation-induced intestinal injury and IBS. Due to the increasing number of cancer patients, radiation-induced intestinal injury is becoming a complication of abdominal/pelvic tumor patients undergoing radiation therapy, seriously affecting patient prognosis and quality of life. Therefore, the role of lncRNAs in radiation-induced intestinal injury is also worthy of attention. In addition, currently, lncRNA-targeted therapy is mostly aimed at cancer treatment and is primarily confined to preclinical studies. Therefore, there is still a long way to go in clinical studies of lncRNA. Nevertheless, it is essential to understand which lncRNAs change expression, which lncRNAs play a role, and how they play their role in diseases. It is possible to determine the progression of diseases by detecting the expression of lncRNAs or by targeting the expression of lncRNAs in treatment, which may provide a new method for the treatment of intestinal diseases and other systemic diseases related to the intestinal barrier, such as diabetes, nervous system diseases, cardiovascular diseases, and immune system diseases.

In conclusion, this article reviews the emerging role of lncRNAs in the intestinal barrier and intestinal diseases. As there are currently no FDA-approved therapies targeting intestinal epithelial barriers, lncRNAs may serve as a potential diagnostic and therapeutic strategy for treating intestinal barrier damage-associated diseases.

## Figures and Tables

**Figure 1 biomolecules-13-01674-f001:**
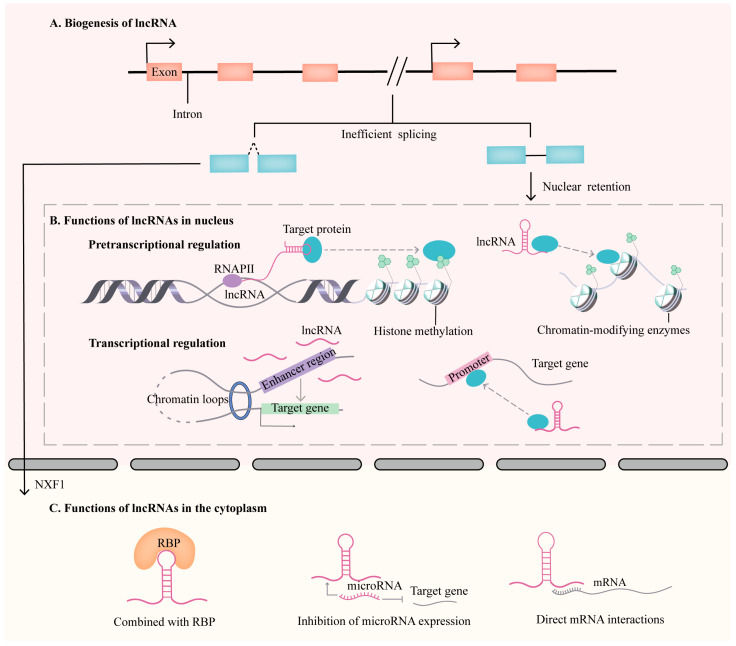
The biogenesis and function of lncRNAs. (**A**) Biogenesis of lncRNA: Lncrnas are weakly transcriptionally spliced, resulting in partial retention in the nucleus. Most lncRNAs are exported to the cytoplasm via NXF1. (**B**) Functions of lncRNAs in the nucleus: In the nucleus, lncRNAs interact with DNA to form an RNA/DNA hybrid or interact with chromatin-modifying enzymes, thereby affecting the chromatin modification state. LncRNAs also regulate the functions of promoters and enhancers of target genes, resulting in the enhancement or inhibition of the expression of parental genes or other genes. (**C**) Functions of lncRNAs in the cytoplasm. In the cytoplasm, lncRNAs can participate in posttranscriptional regulation by interacting with different RBPs, microRNAs, and mRNAs. RBP: RNA-binding proteins; RNAPII: RNA polymerase II.

**Figure 2 biomolecules-13-01674-f002:**
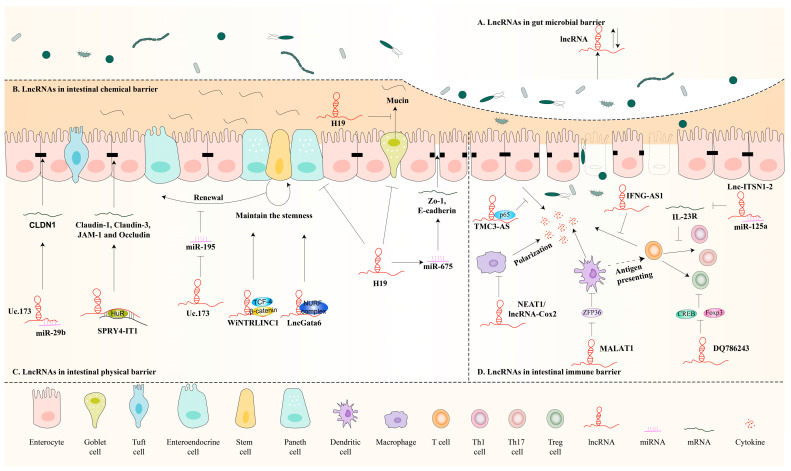
Effects of lncRNAs on the intestinal barrier. (**A**) LncRNAs in gut microbial barrier: The injury factors lead to intestinal dysbiosis, thus affecting the expression of lncRNA. (**B**) LncRNAs in intestinal chemical barrier: LncRNAs (H19) affect the thickness of the intestinal mucus layer by influencing mucin secretion. (**C**) LncRNAs in intestinal physical barrier: On the one hand, lncRNAs can exert an influence on the damage and repair of intestinal stem cells, Paneth cells, goblet cells, and epithelial cells (Uc.173, WiNTRLINC1, LncGata6, H19). On the other hand, lncRNAs can affect cell–cell junctions (Uc.173, SPRY4-1T1). (**D**) LncRNAs in intestinal immune barrier: LncRNAs can regulate the intestinal immune balance by influencing the polarization of macrophages (NEAT1, LncRNA-Cox2), the secretion of cytokines (TMC3-AS1, MALAT1), the proliferation and differentiation of epithelial cells and dendritic cells, and the activation of T cells (IFNG-AS1, Lnc-ITSN1-2, DQ786243).

**Table 2 biomolecules-13-01674-t002:** LncRNAs and IBS.

Long Noncoding RNA	Expression	Models	Functions	Potential mechanism	References
H19	↓	IBS-D patient’s colon	intestinal barrier	AQP1 and AQP3	[108]
XIST	↑	NIH mice	apoptosis and inflammatory response	5-TH and SERT	[109]

SERT: serotonin reuptake transporter; 5-HT: 5-hydroxytryptamine. “↑” arrows represent expressions going up, “↓” arrows represents expressions going down.

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
