# Peer review of "The Role of Long Noncoding RNAs in Intestinal Health and Diseases: A Focus on the Intestinal Barrier"

_biomolecules, 2023, doi:10.3390/biom13111674_

Round 1

Reviewer 1 Report

Comments and Suggestions for Authors

The manuscript by Qianying Lu et.al, demonstrated lncRNAs are essential in the development and regulation of a variety of biological processes and diseases. It is involved in intestinal barrier function and homeostasis maintenance. The manuscript reviews the composition and function of intestinal barrier and the emerging role of lncRNAs in intestinal barrier from four aspects: chemical barrier, physical barrier, immune barrier and microbial barrier. It explores potential lncRNAs that involved in intestinal barrier and provides new ideas for the diagnosis and treatment of intestinal barrier damage associated diseases in the clinic.

I think the manuscript accept after minor revision.

1.      The logic of this manuscript needs further improvement

Comments on the Quality of English Language

1.      The language of paper need to be touch up.

Author Response

Dear reviewer 1:

Thank you for your valuable comments on our manuscript titled " LncRNAs in Intestinal Health and Disease: Focusing on Intestinal Barrier" (biomolecules-2613576). All these comments are valuable and helpful for us to revise and improve our paper, as well as important guidance for our research. We have carefully studied these comments and made revisions in the hope of receiving your approval.

1. The logic of this manuscript needs further improvement

The author’s response

Thank you for your valuable comments. Our previous manuscript consists of four parts: Composition and function of intestinal barrier; Biogenesis and regulatory functions of lncRNAs; The emerging role of lncRNAs in intestinal barrier; The role of lncRNAs in intestinal diseases. We read through the full manuscript, integrated the first part into the third part in the revised version, and checked and modified the full manuscript in order to make the article more logical. Please check.

2. The language of paper need to be touch up.

The author’s response

Thank you for your comments. We have submitted our manuscript to MDPI English Editing Department for English editing, and the editing certificate is as follows. Please check.

Reviewer 2 Report

Comments and Suggestions for Authors

This review is looking to explore the role of lncRNA in the intestinal barrier and hopefully provide some potential targets for future therapeutics. Although I think the attempted organization into the four aspects of the intestinal barrier is a good idea, I think the execution and final product have some serious flaws making the information in this review difficult to trust.

Is this review meant to be focused on humans? All mammalian organisms? Even by looking at your references, it is unclear. Please specify in the abstract and title when you decide.

I detected inappropriate referencing of papers and figures.  This seems to be a major issue for a Review manuscript.

Examples include reference 14, line 56. You use this referenced review as a reference supporting higher tissue specificity of lncRNA.  First off, higher than what? Second, this reference is a review showing no data that lncRNA are more tissue-specific than any other gene species.

The use of Figure 1: It is not explained well, the figure caption should state what acronyms such as AMPs or SIgA.  Peyer's patches are proportionally not correct (in their size relative to crypt) and where is this in the intestine? Was Biorender used to make this figure? If it was, state it. The same applies to the rest of the figures.  Figure 3 is even more disarray with a lot going on and little explanation.

There are some sentences that I don't believe to be scientifically sound:

line 53: not all lncRNA have polyA tails and 5'caps?

line 85, it looks like there was a mistaken pasting of the protein name "Neat1" at the beginning of the sentence.  If not, I don't understand what Neat1 has to do with the rest of the sentence.

The definition of lncRNA being more than 200 nt is also repeated too often throughout the review. The exact sentence referenced above about being similar to mRNA is repeated (almost copied and pasted) into lines 155-158.

In it's current form, this manuscript has too many errors to be trusted as a credible source for compiling information into a review.

Comments on the Quality of English Language

First off, there needs to be some hired editorial help to make this review comprehensible.

I don't even know where to begin and what examples to provide because almost every sentence has an error in it. 

Author Response

Dear reviewer 2:

Thank you for your valuable comments on our manuscript titled " LncRNAs in Intestinal Health and Disease: Focusing on Intestinal Barrier" (biomolecules-2613576). All these comments are valuable and helpful for us to revise and improve our paper, as well as important guidance for our research. We have carefully studied these comments and made revisions in the hope of receiving your approval.

1. Is this review meant to be focused on humans? All mammalian organisms? Even by looking at your references, it is unclear. Please specify in the abstract and title when you decide.

The author’s response

Thank you for your valuable comments. In this paper, we reviewed the literature on cells, animal models and clinical patients to find potential lncRNAs, so as to provide a new research basis for the clinical treatment of intestinal diseases. We have specified in the abstract, please check.

2. I detected inappropriate referencing of papers and figures. This seems to be a major issue for a Review manuscript.

Examples include reference 14, line 56. You use this referenced review as a reference supporting higher tissue specificity of lncRNA.  First off, higher than what? Second, this reference is a review showing no data that lncRNA are more tissue-specific than any other gene species.

The author’s response

Thank you for your valuable comments. We checked all the references and made changes to some inappropriate citations. As for the reference 14 in the previous manuscript that you mentioned, we have reviewed and cited the new references (reference 15-16), and replaced the word “higher” with “high” in the manuscript. Please check.

3. The use of Figure 1: It is not explained well, the figure caption should state what acronyms such as AMPs or SIgA. Peyer's patches are proportionally not correct (in their size relative to crypt) and where is this in the intestine? Was Biorender used to make this figure? If it was, state it. The same applies to the rest of the figures. Figure 3 is even more disarray with a lot going on and little explanation.

The author’s response

Thank you for your valuable comments. We have checked all the Figures carefully and labelled the acronyms in the figure legends.

In Figure 1, Peyer's patches are located in the small intestinal submucosa, containing a large number of lymphocytes and contacting antigenic substances in the intestine via M cells, the location and size of which was indeed an oversight on our manuscript, we apologies for our carelessness. However, we found that Figure 3 basically contains the 4 layers of barriers in Figure 1, so we have fused Figure 1 into Figure 3.

As for Figure 3, we modified the image to make it clearer. We circled lncRNAs that play a role in the same barrier together, and at the same time, modified the figure legends. Figure 3 and the figure notes are shown below:Figure 2. Effects of lncRNAs on the intestinal barrier. A. LncRNAs in gut microbial barrier: The injury factors lead to intestinal dysbiosis, thus affecting the expression of lncRNA. B. LncRNAs in intestinal chemical barrier: LncRNAs (H19) affect the thickness of the intestinal mucus layer by influencing mucin secretion. C. LncRNAs in intestinal physical barrier: On the one hand, lncRNAs can exert an influence on the damage and repair of intestinal stem cells, Paneth cells, goblet cells, and epithelial cells (Uc.173, WiNTRLINC1, LncGata6, H19). On the other hand, lncRNAs can affect cell–cell junctions (Uc.173, SPRY4-1T1). D. LncRNAs in intestinal immune barrier: LncRNAs can regulate the intestinal immune balance by influencing the polarization of macrophages (NEAT1, LncRNA-Cox2), the secretion of cytokines (TMC3-AS1, MALAT1), the proliferation and differentiation of epithelial cells and dendritic cells, and the activation of T cells (IFNG-AS1, Lnc-ITSN1-2, DQ786243).

Moreover, we use Adobe Illustrator software instead of Biorender for figures drawing.

All the changes in the manuscript have been highlighted, please check.

4. line 53: not all lncRNA have polyA tails and 5'caps?

The author’s response

Thank you for your valuable comments. We replaced the sentence “lncRNA have polyA tails and 5'caps” with “most of lncRNAs have 5′-end m7G caps and 3′-end poly(A) tails”, Please see line 63. Please check.

5. line 85, it looks like there was a mistaken pasting of the protein name "Neat1" at the beginning of the sentence. If not, I don't understand what Neat1 has to do with the rest of the sentence.

The author’s response

Thank you for your valuable comments. We apologise for this error due to an oversight on our part, the error has been removed. Please check.

6. The definition of lncRNA being more than 200 nt is also repeated too often throughout the review. The exact sentence referenced above about being similar to mRNA is repeated (almost copied and pasted) into lines 155-158.

The author’s response

Thank you for your valuable comments. We have deleted duplicate sentences in the revised version. Please check.

7. In it's current form, this manuscript has too many errors to be trusted as a credible source for compiling information into a review.

The author’s response

Thank you for your valuable comments. We have examined the entire manuscript and made changes to the text, figures, tables, citations, and references in accordance with your comments. Please check.

8.Comments on the Quality of English Language

First off, there needs to be some hired editorial help to make this review comprehensible.

I don't even know where to begin and what examples to provide because almost every sentence has an error in it. 

The author’s response

Thank you for your comments. We have submitted our manuscript to MDPI English Editing Department for English editing, and the editing certificate is as follows. Please check. 

Reviewer 3 Report

Comments and Suggestions for Authors

Lu et. al. present an intersting review on involvment of lncRNA in intestinal barier. The work seem quite comprehensive and well written. I also have to commend the authors for the pictures 

I have some major points:

Tables 1 and 2 lack footnote with abbreviations description

References seem not formated in MDPI style.

There is no space before the reference in the text.

All the other comments can bi found in attached file.

Comments on the Quality of English Language

English is generally fine, my comments are in the attached pdf.

Author Response

Dear reviewer 3:

Thank you for your valuable comments on our manuscript titled " LncRNAs in Intestinal Health and Disease: Focusing on Intestinal Barrier" (biomolecules-2613576). All these comments are valuable and helpful for us to revise and improve our paper, as well as important guidance for our research. We have carefully studied these comments and made revisions in the hope of receiving your approval.

1.Tables 1 and 2 lack footnote with abbreviations description

The author’s response

Thank you for your comments. The footnotes in Tables 1 and 2, as well as table 3 have been added to the revised manuscript. Please check.

2.References seem not formated in MDPI style.

The author’s response

Thank you for your comments. We have revised the references according to the MDPI reference format, please check.

3. There is no space before the reference in the text.

The author’s response

Thank you for your comments.

We checked all the references in the text, and added spaces before the reference in the text, please check.

4.Comments on the Quality of English Language: English is generally fine, my comments are in the attached pdf.

The author’s response

Thank you for your careful and friendly comments, which are of great help to our manuscript. We have read your amendments and comments carefully, and have corrected the grammar problems according to your comments. For other questions, the replies are as follows:

  • line 257: These two sentences have too many repetitive words.

The author’s response

Thank you for your comments. We have made changes, as detailed in the revised version of the line 198. Please check.

  • line 207: Check the numenclature for this specifi miRNA and correct.

The author’s response

Thank you for your comments. We have modified the miRNA in the revised manuscript, as detailed in the revised version of the line 211. Please check.

  • line 301: Explain FA

The author’s response

Thank you for your comments. The explanation of FA has been explained specifically in the manuscript line 260. Please check.

  • line 388: Explain the abbreviation ICC

The author’s response

Thank you for your comments. The relevant content here has been removed as we discovered that the cited article was retracted. Please check.

  • Line 372: Revise the nomenclature of this miRNA

The author’s response

Thank you for your comments. We have modified the miRNA in the revised manuscript, as detailed in the revised version of the line 340. Please check.

  • line 419: If this is gene you should use proper nomenclature. For genes we use capital letters

The author’s response

Thank you for your comments. We apologize for this error, and the gene has been corrected in the revised version, as detailed in line 390. Please check.

  • Table3: Revise the whole table according to my suggestion or your own. It needs to be the same in every line

The author’s response

Thank you for your comments. We checked Tables 1- 3 and changed all initials in the table to lower case, so that the format in all the tables was uniform. In addition, we modified the table according to the order of lncRNAs in the article. Please check.